# Differential Roles of Dendritic Cells in Expanding CD4 T Cells in Sepsis

**DOI:** 10.3390/biomedicines7030052

**Published:** 2019-07-18

**Authors:** Samuel Darkwah, Nodoka Nago, Michael G. Appiah, Phyoe Kyawe Myint, Eiji Kawamoto, Motomu Shimaoka, Eun Jeong Park

**Affiliations:** 1Department of Molecular Pathobiology and Cell Adhesion Biology, Mie University Graduate School of Medicine, Tsu, Mie 514-8507, Japan; 2Department of Clinical Nutrition, Suzuka University of Medical Science, Suzuka, Mie 510-0293, Japan; 3Department of Emergency and Disaster Medicine, Mie University Graduate School of Medicine, Tsu, Mie 514-8507, Japan

**Keywords:** sepsis, spleen, mesenteric lymph nodes, dendritic cells, CD4 T cells, IL-1β

## Abstract

Sepsis is a systemically dysregulated inflammatory syndrome, in which dendritic cells (DCs) play a critical role in coordinating aberrant immunity. The aim of this study is to shed light on the differential roles played by systemic versus mucosal DCs in regulating immune responses in sepsis. We identified a differential impact of the systemic and mucosal DCs on proliferating allogenic CD4 T cells in a mouse model of sepsis. Despite the fact that the frequency of CD4 T cells was reduced in septic mice, septic mesenteric lymph node (MLN) DCs proved superior to septic spleen (SP) DCs in expanding allogeneic CD4 T cells. Moreover, septic MLN DCs markedly augmented the surface expression of MHC class II and CD40, as well as the messaging of interleukin-1β (IL-1β). Interestingly, IL-1β-treated CD4 T cells expanded in a dose-dependent manner, suggesting that this cytokine acts as a key mediator of MLN DCs in promoting septic inflammation. Thus, mucosal and systemic DCs were found to be functionally different in the way CD4 T cells respond during sepsis. Our study provides a molecular basis for DC activity, which can be differential in nature depending on location, whereby it induces septic inflammation or immune-paralysis.

## 1. Introduction

Sepsis is a major healthcare concern with a high rate of mortality stemming from the failure of host immunity to protect against polymicrobial invasions [1,2]. The innate and adaptive immune responses observed in sepsis usually occur in biphasic steps; in particular, the early acute phase of immunologic hyperactivation and the late chronic phase extending to persistent immunosuppression [3,4,5]. The immunologic hyperactivation that occurs in sepsis frequently leads to widespread inflammation, multiple organ failure, and mortality during the early phase via the systemic release of vast quantities of pro-inflammatory cytokines, a phenomenon known as a cytokine storm [2,6]. By contrast, persistent immunosuppression or immune-paralysis during sepsis induces aberrant effects on immune cells, bringing about secondary infections and eventually death [5,7]. Although improvements in intensive care have achieved remarkable gains in decreasing the frequency of septic mortality, severe problems encountered in a majority of septic survivors, as well as early victims of the condition, appear to originate from both the impaired innate and adaptive immune systems.

Dendritic cells (DCs), which are representative antigen-presenting cells for CD4 T cells act, not only as sentinels for initiating adaptive immune responses, but also as cardinal regulators of innate immunity [8]. DCs originate from bone marrow hematopoietic stem cells [9]. Heterogeneous in nature, DCs and their subsets can be divided into myeloid and plasmacytoid DCs depending on their functions and markers [10,11]. Distinct patterns in the intratissue migration of DCs to T-cell zones depend on their differential subsets [12], although both tissue-resident and intertissue-migrated DCs are thought to cooperate in order to prime CD4 T cells [13]. Thus, the migration of DCs within and to tissues, in order to translocate antigens and accurately position DCs inside target tissues, are prerequisite steps for launching tissue-specific immune responses [14,15].

Emerging evidence suggests that both the function and phenotype of DCs are altered by sepsis in humans [16,17] and mice [18,19,20,21]. Because sepsis-induced functional alterations of DCs modify immune responses, understanding the precise role played by sepsis-affected DCs is important. Although DCs are known to control immune responses, little is known about the functional differences between the DCs from systemic (e.g., SP) versus mucosal (e.g., mesenteric lymph node (MLN)) tissues in sepsis. By studying the compartmental distinctiveness of septic DCs in exerting different inflammatory responses, we can learn about the dynamics underlying the DC-mediated spatiotemporal balance and surveillance, which occurs between inflammation and immune suppression during sepsis.

Human primary DCs have been isolated, mostly from the blood samples of sepsis patients, to the greatest extent possible [22,23]. Cecal ligation and puncture (CLP) is a commonly used method to induce experimental polymicrobial sepsis in mice in order to mimic human sepsis [24,25], although this model is susceptible to age and strain variability [26]. Thus, CLP is regarded as an alternative model for closely examining the function of septic DCs isolated from different tissues. One example of measuring either pro- or anti-inflammatory effects of DCs involves a proliferation assay for CD4 T cells co-cultured with the DCs. This is done under conditions of a mixed lymphocyte reaction (MLR), which allows allogeneic DCs to stimulate lymphocytes in vitro or ex vivo [27]. This assay could prove instrumental in testing the differential roles played by systemic, versus mucosal, DCs in modifying immune responses in the CLP sepsis model. 

It is intriguing to note that mucosal DCs from septic mice display a differential role in the activation or proliferation of CD4 T cells, compared to systemic DCs from the same mice, possibly due to a pro-inflammatory mediator preferentially released from mucosal DCs. CD4 T-cell proliferation occurs, at least in part, via the action of DC-released IL-1β [28]. Here, we found that MLN DCs exhibited greater proliferation of allogeneic CD4 T cells compared to SP DCs in CLP mice. Intriguingly, IL-1β messaging in MLN DCs from CLP mice was shown to be markedly increased. Indeed, CD4 T cells were highly proliferated in culture, when treated with exogenous IL-1β. This may imply that MLN DCs of septic mice may upregulate T-cell activation and proliferation by secreting IL-1β. Thus, our study proposes that DCs, depending on their compartments, respond differentially to allogeneic CD4 T cells during sepsis.

## 2. Results

### 2.1. CD4 T-cell Proliferation is Augmented by Allogeneic MLN DCs in Sepsis

CD4 T cells are the lymphocyte subset that orchestrate proper innate and adaptive immune responses [29]. Sepsis often accompanies lymphopenia [30,31]. We first examined changes in cellularity in systemic and mucosal lymphoid organs in a sepsis mouse model using cecal ligation and puncture (CLP). The ratio of CD4 T cells to mononuclear cells in MLN and peripheral (inguinal) LNs (PLN) was diminished by sepsis, whereas the quantity of cells in SP or Peyer’s patches (PP) did not change significantly in either group of mice (Figure 1A). Unlike systemic tissues, such as SP and PLN (inguinal LN), the mucosal lymphoid organs (MLN and PP) showed an increase in DCs in sepsis (Figure 1B). B cells were slightly, though still significantly, increased only in MLNs (Figure 1C). Thus, sepsis-induced lymphopenia was, in fact, restricted to CD4 T cells in lymph nodes under our experimental conditions of CLP. The representative plots of flow cytometry analysis by which the Figure 1 graphs are made, are shown in Appendix A. These results indicate that sepsis gives rise to increased mucosal DCs, which may play a distinct role in modulating septic inflammation, unlike that which is observed with systemic DCs.

To directly determine the impact of mucosal DCs on CD4 T-cell proliferation, we employed a mixed lymphocyte reaction (MLR) [27]. PP is a mucosal lymphoid tissue but known to be prone to sepsis-induced apoptotic cellular loss [32,33]. In accordance with these findings, the CLP mice exhibited a marked reduction in PP size (data not shown), which made it difficult to separate the DCs enough to perform the analysis. Thus, MLN was used to provide mucosal DCs in the current analyses. CD4 T cells (from the SP of Balb/c mice) were fluorescently labeled with CFSE and then co-cultured with DCs (from SP or MLN of C57BL/6J mice) at 4:1 (T/DC ratio) for 7 days. Proliferation of the CD4 T cells co-cultured with MLN DCs of CLP (+) was significantly increased compared with those of CLP (−), as well as with SP DCs of CLP (+) mice, as shown in the representative histograms (Figure 2A) and bar graphs (Figure 2B). However, there was no significant difference in the higher proliferation levels observed in SP DCs under conditions of CLP (+) compared to those of CLP (−) mice (Figure 2). Instead, their proliferating effect on CD4 T cells exhibited the same trend observed with a reduction brought about by co-cultured septic SP DCs, although statistical significance was not reached. Therefore, these results suggest that mucosal DCs tend to facilitate allogeneic CD4 T cells during the 24 h following CLP, a pattern which is quite different from that displayed by systemic DCs. This may be indicative of compartmental differences in DC activity upon the onset of septic inflammation.

### 2.2. Some Activation Markers Are Highly Increased in MLN DCs in Sepsis

In order to identify any plausible mechanism used by DCs that might affect CD4 T-cell proliferation, we next examined the change in surface markers between DCs from the SP and MLNs of CLP (−) and (+) mice. We isolated the total mononuclear cells (MNCs) from both tissues and performed flow cytometry analysis to determine the differential level of APC markers on CD11c^+^ cells, including major histocompatibility complex (MHC) class II, CD40, CD80, or CD86 (Figure 3A). Both SP and MLN cells appeared to exhibit a slight increase in the expression on CD11c^+^ cells during sepsis, as shown in the histograms of Figure 3A. MHC class II has been described as a key molecule used by DCs to activate antigen-specific CD4 T cells [34]. The level of MHC class II expression on MLN DCs was significantly higher compared to that on SP DCs under both healthy and septic conditions (Figure 3B). Moreover, the MHC class II molecules, expressed on MLN DCs, were augmented by sepsis (Figure 3B). CD40 is known to engage in the activation of both DCs and CD4 T cells [35]. The MLN DCs of septic mice exhibited the highest level of CD40 expression, although the expression of CD40 on SP DCs also increased significantly under conditions of sepsis (Figure 3B). On the other hand, the expressions of CD80 on both SP DCs and MLN DCs, and those of CD86 on SP DCs, were shown to be increased by sepsis; however, neither marker was significantly higher on MLN DCs compared to SP DCs in septic mice (Figure 3B). These results imply that the MHC class II and CD40 molecules of mucosal DCs may, at least partly, contribute to CD4 T-cell proliferation during sepsis.

The monocyte-derived inflammatory DCs are expected to upregulate MHC class II and be functional in antigen presentation [36,37]. We examined the expression of the MHC class II on CD3ε^-^B220^-^Gr-1^-^TER119^-^CD11c^+^ DCs of SP and MLN in the context of current sepsis model using flow cytometry (Appendix A). The DCs of both organs from CLP (-) and CLP (+) mice showed an increase in expression of MHC class II in sepsis (Appendix A). Thus, these results support a possibility that both SP DCs and MLN DCs could have been derived, at least partly, from monocytes during sepsis. We next tested the expression level of other inflammatory markers on the DCs from SP and MLN in sepsis. The cells were prepared from SP or MLN of three mice per group with either CLP (−) or CLP (+) and stained with mAbs to CD3ε, B220, Gr-1, TER119, CD11c, and MHC class II, along with inflammatory DC markers including F4/80, CD11b, CD107b, FcεR1α, CD206, or Ly6c [38,39,40]. Both total CD3ε ^-^B220^-^Gr-1^-^TER119^-^ cells and CD3ε^-^B220^-^Gr-1^-^TER119^-^CD11c^+^MHC class II^+^ cells were subjected to analyzing expression of each marker (Appendix A). Although the segregated subsets of the DCs were not shown in the dot plot, as previously reported [38,41], we found that the CD3ε^-^B220^-^Gr-1^-^TER119^-^CD11c^+^MHC class II^+^ cells from both SP and MLN showed an increase in expressing F4/80, CD107, CD107b, FcεR1α, and CD206, compared to total CD3ε^-^B220^-^Gr-1^-^TER119^-^ cells (Appendix A). It may imply that the CD3ε^-^B220^-^Gr-1^-^TER119^-^CD11c^+^MHC class II^+^ cells represent DCs which have potential to be inflammatory in sepsis. Actually, the expressions of the F4/80, CD11b, CD107b, and FcεR1α on SP DCs, and CD11b on MLN DCs slightly increased during sepsis (Appendix A). Similarly, percentage of the cells that express those inflammatory DC markers (F4/80, CD11b, CD107b, FcεR1α, and Ly6c on SP DCs, and CD11b on MLN DCs) in CD3ε^-^B220^-^Gr-1^-^TER119^-^CD11c^+^MHC class II^+^ cells exhibited a sepsis associated increase (Appendix A). Therefore, these results indicate that the CD3ε^-^B220^-^Gr-1^-^TER119^-^CD11c^+^MHC class II^+^ cells in lymphoid organs may possess a potential to induce expression of inflammatory markers and/or thereby differentiate to inflammatory DCs in sepsis.

We also investigated the expression and positive-cell ratio of chemokine receptors CCR2 and CCR7 implicated in inflammatory DC function [42,43,44]. The CD3ε^-^B220^-^Gr-1^-^TER119^-^CD11c^+^MHC class II^+^ cells were shown to highly express both chemokine receptors compared to total CD3ε^-^B220^-^Gr-1^-^TER119^-^ cells, especially in MLN (Appendix A). However, the expression level and positive-cell ratio of CCR2 and CCR7 among CD3ε^-^B220^-^Gr-1^-^TER119^-^CD11c^+^MHC class II^+^ cells did not exhibit any sepsis-induced remarkable alteration (Appendix A).

### 2.3. Level of IL-1β mRNA is Markedly Increased in Septic MLN DCs

To determine whether septic DCs can produce pro-inflammatory mediators in a differential manner, depending on the compartment, we next analyzed the mRNA levels of cytokines including IFN-γ, TNF-α, IL-1β, and IL-6 expressed in DCs from the SP and MLN of CLP (−) and (+) mice. We found that the mRNA expressions of TNF-α, IL-1β, and IL-6 were upregulated in MLN DCs and those of IL-1β and IL-6 were augmented in SP DCs by sepsis (Figure 4). The mRNAs of TNF-α were decreased by sepsis in SP DCs (Figure 4). Importantly, *IL*-*1β* levels of MLN DCs were markedly elevated by sepsis; in fact, *IL*-*1β* expression of septic MLN DCs overwhelmed that of septic SP DCs (Figure 4). Therefore, these results suggest that *IL*-*1β* expression of MLN DCs is positively controlled by sepsis, and that the IL-1β generated by septic mucosal DCs might play a role in modulating antigen-dependent lymphocyte function.

### 2.4. CD4 T-Cell Proliferation is Augmented by IL-1β Treatment in a Dose-Dependent Manner

It has been previously reported that IL-1β augments CD4 T-cell expansion in an antigen-dependent fashion [28]. Based on our results (Figure 4), it is tempting to speculate that the IL-1β upregulated by mucosal DCs may engage in proliferating CD4 T cells during sepsis. This has led us to examine whether or not exogenous treatment with IL-1β augments CD4 T-cell proliferation. We treated CFSE-labeled CD4 T cells with recombinant mouse IL-1β (rmIL-1β) and cultured them for 3 days. The cytokine concentration was chosen based on the fact that ED_50_ of rmIL-1β for lymphocyte proliferation is 2–10 pg/mL [45]. Cell proliferation with 2, 5, or 10 pg/mL increased in a dose-dependent manner, whereas the proliferation ratio plateaued at 2, 5, or 10 ng/mL without any remarkable change, as shown in representative flow-cytometry histograms (Figure 5A), as well as line graph (Figure 5B).

In order to examine if this increase in CD4 T-cell proliferation is specific to IL-1β, we tested an effect of TNF-α and found that there was no significant augmentation with TNF-α, even at 100 ng/mL of its concentration (Appendix A). These results support our demonstration that IL-1β generated from septic mucosal DCs facilitates the proliferation of CD4 T cells. Because regulatory T cells are thought to involve in immuno-paralysis in the late phase of sepsis [46], we have attempted to measure expression of FoxP3 in the CD4 T cells proliferated or skewed by IL-1β. The CD4 T cells stained with CFSE were cultured with either 10 pg/mL or 10 ng/mL of IL-1β for 3 days and analyzed for FoxP3 expression in CFSE-diluted cells. We found that FoxP3 was more expressed by treatment with high (10 ng/mL) than low dose (10 pg/mL) of IL-1β (Appendix A), raising a possibility of IL-1β induced CD4 T-cell differentiation to regulatory T cells in sepsis.

Based on our results of the possible role played by mucosal DC-producing IL-1β during sepsis, we proposed a model to illustrate the compartmental differences DCs in systemic SP and mucosal MLN organs exert vis-à-vis CD4 T-cell proliferation. This process is depicted in Figure 6, in which MLN DCs produce more IL-1β to more effectively expand allogeneic CD4 T cells.

## 3. Discussions

In the present study, we examined the ability of DCs to elicit an adaptive immune response in the context of MLR, in which mucosal MLN DCs induced a significant increase in CD4 T-cell proliferation in a septic environment. MLN DCs proved superior to systemic SP DCs in increasing CD4 T-cell proliferation in our experimental sepsis model. Both SP DCs and MLN DCs increased the activation markers, such as MHC class II, CD80, or CD86 during sepsis. The expression of some surface markers (e.g., MHC class II or CD40) increased on septic MLN DCs, indicating that these molecules engage in functionally activating mucosal DCs during sepsis. Furthermore, mucosal DCs in sepsis preferentially expressed IL-1β messaging, which has been reported to directly expand CD4 T cells [28]. We subsequently confirmed that the proliferation rate of CD4 T cells increased, in a dose-dependent manner, when exposed to exogenously treated IL-1β. Thus, our findings suggest a model that sepsis precipitates the formation of a mucosal lymphoid niche favorable for DCs to facilitate CD4 T-cell proliferation and/or activation. Moreover, we contend that this occurs in a more optimal manner than that achieved by systemic DCs, due to the action of key DC-produced mediators, such as IL-1β.

The mechanism by which septic conditions upregulate IL-1β in mucosal DCs remains unclear. However, one plausible explanation can be put forward. Toll-like receptor 4 (TLR4) activation turned out to induce IL-1β production in the bronchoalveolar lavage fluid of mice that inhaled lipopolysaccharide (LPS) [47]. The same mechanism was also implicated in thioglycollate-elicited peritoneal macrophages in tandem with high mobility group box-1 (HMGB1) [48]. In addition, IL-1β was reported to play a modulating role in the TLR4-dependent production of other pro-inflammatory cytokines in pathogenically challenged human epithelial cells [49]. Based on these previous findings, the hypothesis that TLR4 signaling is required to trigger the release of IL-1β under conditions of systemic inflammation appears quite plausible, although TLR4 on DCs are known to regulate inflammatory neutrophils or spleen DCs during sepsis [50,51]. Thus, it can be deduced that the inflammatory action of MLN DCs in sepsis, via IL-1β, is dependent upon mediation via TLR4 and/or HMGB1. Finally, in the near future it would be worthwhile to elucidate the precise roles of these two mucosal-DC molecules in regulating the progression of sepsis.

In agreement with the aforementioned findings on the positive effect of mucosal DC-generated IL-1β on the immune response during sepsis, it is worth discussing two additional aspects. First, in addition to the profoundly elevated mRNA levels of IL-1β, a significant reduction in the mRNA level of the immune checkpoint ligand PD-L2 might yield an additive effect to MLN DCs in promoting immune responses, such as CD4 T-cell proliferation (data not shown). Unlike MLN DCs, which display such a capacity in proliferating CD4 T cells, SP DCs exert an immuno-paralyzing effect in the current CLP model. This immunosuppressive feature of SP DCs, which results from septic conditions (Figure 2), is reflected in their reduced expression of MHC class II and TNF-α in SP DCs (Figure 3A and Figure 4).

MLR is a model that has been primarily used in studying allograft rejection [52,53,54,55,56]. This model is a conventional in vitro method to analyze the levels of activation or proliferation of T cells via co-culturing with allogeneic stimulator cells [27,57]. Because DCs are functionally modified by sepsis [16,17,18,19,20,21], we have incorporated the MLR into the current study in order to determine if septic DCs differentially modulate T-cell function in vitro.

There are a few animal models capable of mimicking human sepsis including lipopolysaccharide (LPS) treatment, peritoneal contamination, and/or CLP [25,26,58]. Of these, the treatment of LPS is a well-established approach to address the role of DCs in septic syndromes. The loss of systemic DCs was revealed to represent a prerequisite event for LPS-induced immunosuppression and mortality in mice [59]. Glucocorticoids were known to exhibit efficacy in ameliorating LPS-induced sepsis by decreasing IL-12 production of SP DCs [60]. Even though systemic DCs were found to relieve endotoxic shock, as shown in these previous studies, the impact of mucosal DCs on LPS-induced sepsis remains to be determined.

The location of DC subsets determines their function; hence, an understanding of the functional heterogeneity innate to tissue DCs is needed if their precise role in septic development is to be demonstrated [61,62]. In the current model, DCs were isolated from SP and MLN 24 hours after CLP, which means that they may be present during the early phase of sepsis when hyperinflammatory responses surge [4]. The DC ratio in the mucosal lymphoid tissues of CLP mice increased, unlike that in systemic tissues, in which no significant differences between CLP (−) and (+) mice were apparent (Figure 1B). The loss of circulating DCs in septic patients, as described above, is likely due to the fact that these clinical samples were obtained from “septic survivors”—i.e., patients who experienced persistent immunosuppression during the late phase [4]. Hence, this could explain the variabilities in DC ratios recorded during sepsis, as they are dependent upon both time and location. Apart from this interpretation, one must also be aware that the CLP model is probably susceptible to variations stemming from age and/or strain [26]. Lymphocytes undergo apoptosis even during the early stage of sepsis [63], and thus lymphopenia in CD4 T cells was also observed in lymphoid organs, as shown in our CLP model (Figure 1A). Nonetheless, the ability of mucosal DCs to proliferate CD4 T cells has been detected in an MLR setting (Figure 2). This would indicate that such DCs have some inflammatory potential, one exerted in an antigen-dependent manner. 

Systemic DCs did not reveal any such ability at this 24-h time point in the CLP model; rather, they seemed to even lose this ability in the immune response to allogeneic CD4 T cells after the onset of sepsis, although the decrease did not reach statistical significance (Figure 2). Based on these results, it can be also be deduced that the functionality differed between systemic and mucosal DCs, and was manifested as the change in CD4 T-cell responses during this particular 24-hour period of sepsis. Although it is believed that immune responses mounted by systemic DCs during sepsis may precede those of mucosal DCs, the possibility that this occurs in reverse order cannot be ruled out. This would signify a gut-originated stream of bacterial translocation to the systemic circulation [64]. Another possibility concerns the differences between systemic and mucosal DCs in inducing CD4 T-cell responses during sepsis, which could be caused by the transition of the intestinal microbiome to a “pathobiome”. This is suggested by the presence of a sepsis-induced polymicrobial burden [65]. DCs are educated, to varying degrees, by the sepsis-induced reshaping of pathogenic microbiota, thus leading to differential effects on inflammation. Nonetheless, the precise factors that allow systemic DCs to exhibit immuno-paralysis in sepsis remain unclear. 

The cells referred to as DCs, and used for stimulating CD4 T cells and evaluating mRNAs expressions in this study, were CD11c^+^ cells obtained from single-cell suspensions of spleens and lymph nodes by using CD11c microbeads. Thus, it may be questionable to designate these cells as “DCs” prior to further dissecting them into several subsets. Nonetheless, those CD11c^+^ cells isolated by the same method as ours were called “DCs” and analyzed in many studies [19,50,51,66,67,68]. In addition, some studies have used the term “CD11c^+^ cells” [34,69,70]. As “DCs” may arguably be the correct description based on previous reports, the use of the term “CD11c^+^ cells” might also be deemed acceptable. 

DCs are heterogeneous and can be largely divided into classical DCs (cDCs) and plasmacytoid DCs (pDCs), depending on their function and surface-marker expression [71,72]. The cDCs representing all DCs, except for pDCs, play a specialized role in presenting processed foreign antigens to T lymphocytes, mostly in order to maintain immunological homeostasis [9]. In contrast, pDCs respond to a broad range of immune disorders including infections, autoimmune diseases, and tumors; for instance, pDCs rapidly secrete type I interferon in response to viral infections [73,74,75]. The number and antigen-presenting function of circulating DCs were downregulated in septic patients [16,76,77]. A decrease in the number or functionality of DCs in lymphoid tissues was also reported in CLP mice [66,67,78]. Given the heterogeneity of DC subsets, as well as their multifaceted alterations during sepsis, both the cDCs and pDCs observed in mucosal tissues need to be characterized in parallel in an independent study, because the specific subsets of DCs may contribute to eliciting distinct outcomes in modulating inflammation.

Among the inflammatory DC markers tested in this study, F4/80 was highly increased by SP DCs of CLP (+) mice (Appendix A). Of note, approximately half of CD3ε^-^B220^-^Gr-1^-^TER119^-^CD11c^+^MHC class II^+^ cells was revealed to be F4/80-expressing subsets (Appendix A). A previous study demonstrated a differentiation of splenic inflammatory monocytes into monocyte-derived DCs that express F4/80 during malaria infection [79]. Thus, our study may propose a speculative scenario that some splenic inflammatory monocytes differentiate into monocyte-derived F4/80-expressing CD3ε^-^B220^-^Gr-1^-^TER119^-^CD11c^+^MHC class II^+^ DCs during sepsis, although the role of the DC subset in modulating inflammatory progression remains to be elucidated in near future.

It would be worthwhile to carry out a comprehensive and valid demonstration of the DC-mediated spatiotemporal differences between systemic and mucosal lymphoid organs in eliciting immune responses during sepsis. Further investigations need to be done under various conditions, including varying levels of severity (using different positions of cecal ligation) or duration (e.g., 6 h, 12 h, 3 days or 7 days) to gain a more comprehensive overview in the context of a sepsis model. The present study provides evidence that DCs, depending on their location, possess a differential capability of regulating inflammation during sepsis.

In conclusion, our study demonstrated that mucosal DCs were superior to systemic DCs in proliferating allogeneic CD4 T cells in sepsis. The septic mucosal DCs markedly augmented the surface expression of MHC class II and CD40, as well as the messaging of IL-1β. Exogenous IL-1β was effective in CD4 T-cell expansion in a dose-dependent manner, implying that this cytokine, released from MLN DCs during sepsis, may play a role, at least partly, in promoting septic inflammation. The present work provides a molecular basis for DC activity, which can be differential in nature depending on location, whereby it induces septic inflammation or immuno-paralysis.

## 4. Methods

### 4.1. Mice

The two species of mice, C57BL/6J and Balb/c (10–13-weeks old), were obtained from Japan SLC (Shizuoka, Japan) and maintained with a 12-h light and 12-h dark cycle at the Experimental Animal facility of Mie University. This study was approved by the Ethics Review Committee for Animal Experimentation of Mie University (approval number: #27-6-2-1-2 approved on 16 November 2018).

### 4.2. Polymicrobial Sepsis

Sepsis was induced in C57BL/6J mice by cecal ligation and puncture as previously described with minor modifications [80]. In brief, under isoflurane anaesthetization, an 18-gauge needle (outer caliber: 1.27 mm) was inserted into the opposite edge of the ileocecal valve of the cecum to make a single puncture, which was then tied at 25–30% of the entire cecal length. This ligation length is a factor for determining septic severity (for instance, 100%, severe; 20%, mild or moderate) [81]. A small amount of intestinal contents was allowed to extrude to ensure their release. After suture surgery, 1 mL of 0.9% NaCl was subcutaneously injected for fluid replenishment. Organ harvest for experiments was done at 24 h after the surgery. On the other hand, under these conditions of CLP, a mortality rate of 60% was obtained when the mice were observed up to 14 days after abdominal surgery (data not shown), which was similar to those reported in previous studies [82,83].

### 4.3. Isolation of CD4^+^ T Cells and CD11c^+^ Dendritic Cells (DCs)

Total mononuclear cells were isolated from lymphoid tissues as described previously [84]. CD4^+^ T cells were isolated from a single-cell suspension via negative selection using a CD4^+^ T-cell Isolation Kit (Miltenyi Biotec, Auburn, CA, USA) according to the manufacturer’s instructions. CD11C^+^ cells were also isolated from a single-cell suspension by using CD11c Microbeads (Miltenyi Biotec) 24 hours after suture surgery for CLP.

### 4.4. Mixed Lymphocyte Reaction (MLR) Culture and CD4 T-Cell Proliferation Assay

CD4^+^ T cells isolated from the spleens of Balb/c mice were labeled with 5 µM 5-(and -6)-carboxyfluorescein diacetate succinimidyl ester (CFSE; Thermo Fisher Scientific, Waltham, MA, USA) according to the manufacturer’s instructions. The labeled cells were co-cultured with allogeneic CD11c^+^ DCs from either the spleen or MLNs of C57BL/6J mice (− or +CLP) in the context of MLR as described previously with some modifications [27,85]. In brief, the CFSE-labeled CD4^+^ T cells (1 × 10^5^ per mL) of Balb/c and CD11c^+^ cells (2.5 × 10^4^ per mL) from C57BL/6J mice were co-cultured for 7 days in RPMI-1640 media (nacalai, Kyoto, Japan) supplemented with 10% FBS (Equitech-Bio, Kerrville, TX, USA), penicillin (100 U/mL)/streptomycin (100 µg/mL) (nacalai), and 50 µM 2-mercaptoethanol in an incubator maintained at 37 °C with 5% CO_2_. Each cell population was determined by the dilution of the CFSE signal in the FL1 channel by flow cytometry using BD C6 Accuri (BD Biosciences, San Jose, CA, USA) [85]. In some experiments, CFSE-labeled CD4 T cells were incubated for 3 days with different concentrations of recombinant mouse IL-1β (R&D Systems, Minneapolis, MN, USA) and recombinant mouse TNF-α (R&D Systems) and their proliferation was analyzed in the same way.

### 4.5. Flow Cytometry Analysis

Mononuclear cells were stained with fluorescently labeled antibodies to CD11c-PE (HL3, BD Biosciences), CD4-FITC (RM4-4, Biolegend, San Diego, CA, USA), B220-biotin (RA3-6B2, Biolegend), B220-APC (RA3-6B2, Biolegend), I-A/I-E-APC (MHC class II, M5/114.15.2, Biolegend), CD40-APC (3/23, Biolegend), CD80-FITC (16-10A1, Biolegend), CD86-FITC (GL-1, Biolegend), CD3ε-biotin (145-2C11), Gr-1-biotin (RB6-8C5, Biolegend), TER119-biotin (TER-119, Biolegend), F4/80-FITC (BM8, Biolegend), CD11b-FITC (M1/70, Biolegend), CD107b-FITC (M3/84, Biolegend), FcεR1α-FITC (MAR-1, Biolegend), CD206-FITC (C068C2, Biolegend), Ly6c-FITC (HK1.4, Biolegend), CCR2-FITC (SA203G11, Biolegend), and CCR7-AlexaFluor488 (4B12, Biolegend). PerCP Streptavidin (Biolegend) was also used to stain the cells treated with biotinylated mAbs. For intracellular staining experiment, FIX & PERM Kit (Thermo Fisher Scientific) was used to stain CD4 T cells with mAb to FoxP3-PE (MF-14, Biolegend). After washing twice with buffer (PBS buffer containing 2% FBS, 2 mM EDTA, and 0.02% sodium azide), the expression level was analyzed by flow cytometry using a BD Accuri C6 cytometer (BD Biosciences). Data analysis was done by using BD Accuri C6 software.

### 4.6. Reverse Transcription (RT) and Quantitative PCR (qPCR)

The total RNA was extracted from CD11c^+^ DCs using Trizol Reagent (Thermo Fisher Scientific). RT was performed with 1 µg of RNA using a PrimeScript RT reagent Kit (Takara Bio, Shiga, Japan) and qPCR was conducted with SYBR Green PCR reagents (Applied Biosystems, Foster City, CA, USA). β-actin was used as an endogenous control to normalize mRNA expression. Gene expression was analyzed by using a StepOnePlus Real-Time PCR System (Applied Biosystems). All reactions were repeated at least twice. The primer sequences were as follows: β-actin, forward 5′- CATCGTACTCCTGCTTGCTG-3′ and reverse 5′-AGCGCAAGTACTCTGTGTGG-3′; IL-1β, forward 5′-GCCTTGGGCCTCAAAGGAAAGAATC-3′ and reverse 5′-GGAAGACACAGATTCCATGGTGAAG-3′; IL-6, forward 5′-TGGAGTCACAGAAGGAGTGGCTAAG-3′ and reverse 5′-TCTGACCACAGTGAGGAATGTCCAC-3′; IFN-γ, forward 5′-CGGCACAGTCATTGAAAGCCTA-3′ and reverse 5′-GTTGCTGATGGCCTGATTGTC-3′; TNF-α, forward 5′-ATAGCTCCCAGAAAAGCAAGC-3′ and reverse 5′-CACCCCGAAGTTCAGTAGACA-3′.

### 4.7. Statistical Analysis

Data are expressed as mean ± standard error of the mean (SEM) for each group. A Student *t* test was used when two groups were compared. A *p* value < 0.05 was considered statistically significant. *, 0.01 < *p* < 0.05; **, 0.001 < *p* < 0.01; ***, *p* < 0.001 between groups. Statistical analysis was done using Microsoft Excel.

## Figures and Tables

**Figure 1 biomedicines-07-00052-f001:**
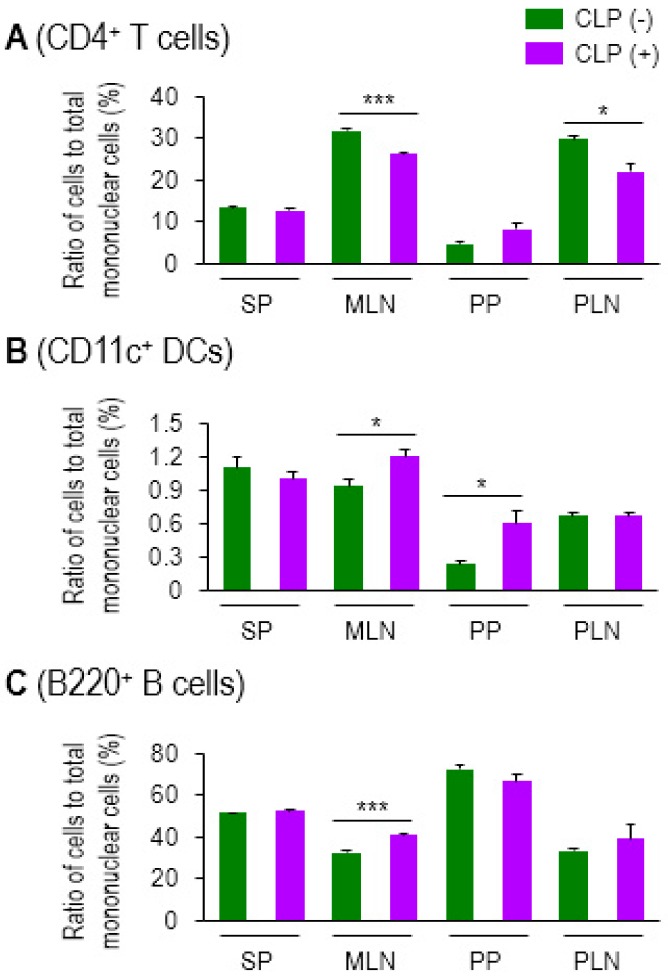
Sepsis induced lymphopenia in CD4 T cells and elevated levels of CD11c^+^ DCs in lymphoid tissues. Ratio of CD4^+^ T cells (**A**), CD11c^+^ DCs (**B**), and B220^+^ B cells (**C**) to total mononuclear cells in spleen (SP), mesenteric lymph nodes (MLNs), Peyer’s patches (PPs), or peripheral lymph nodes (PLNs) was determined by using flow cytometry. For the PLN, inguinal LNs were isolated and used in this assay. CLP (−) indicates control healthy mice and CLP (+) represents CLP mice with moderate sepsis (see Methods section). Bar graphs represent the mean ± SEM obtained from 3 to 4 mice per group. Data are representative of at least three separate experiments. * 0.01 < *p* < 0.05, *** *p* < 0.001.

**Figure 2 biomedicines-07-00052-f002:**
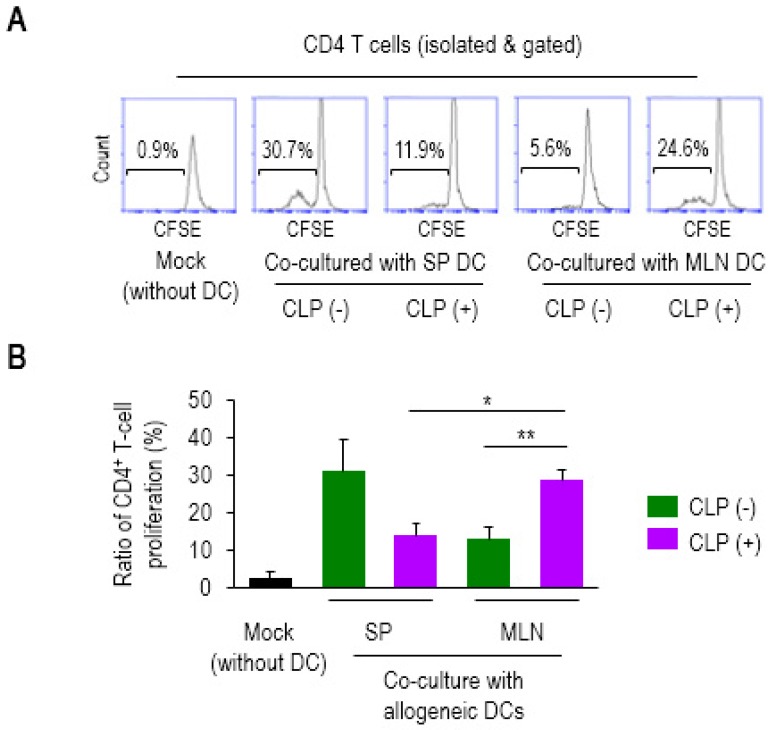
Mesenteric lymph node (MLN) dendritic cells (DCs) in sepsis enhanced the proliferation ratio of allogeneic CD4^+^ T cells. The CD4 T cells were isolated from SP of Balb/c mice, fluorescently labeled with CFSE and co-cultured with mock (without DC), SP (SP DC), or MLN (MLN DC) (isolated from the tissues of C57BL/6J mice) at a ratio of 4:1 for 7 days. The proliferation ratios were determined via measuring diluted fluorescent intensity of a histogram in flow cytometry in which the numbers inside squares represent the percentages of bracketed regions (**A**). Bar graphs represent the mean ± SEM obtained from 4 to 5 mice per group (**B**). Data are representative of at least three independent experiments. * 0.01 < *p* < 0.05, ** 0.01 < *p* < 0.001.

**Figure 3 biomedicines-07-00052-f003:**
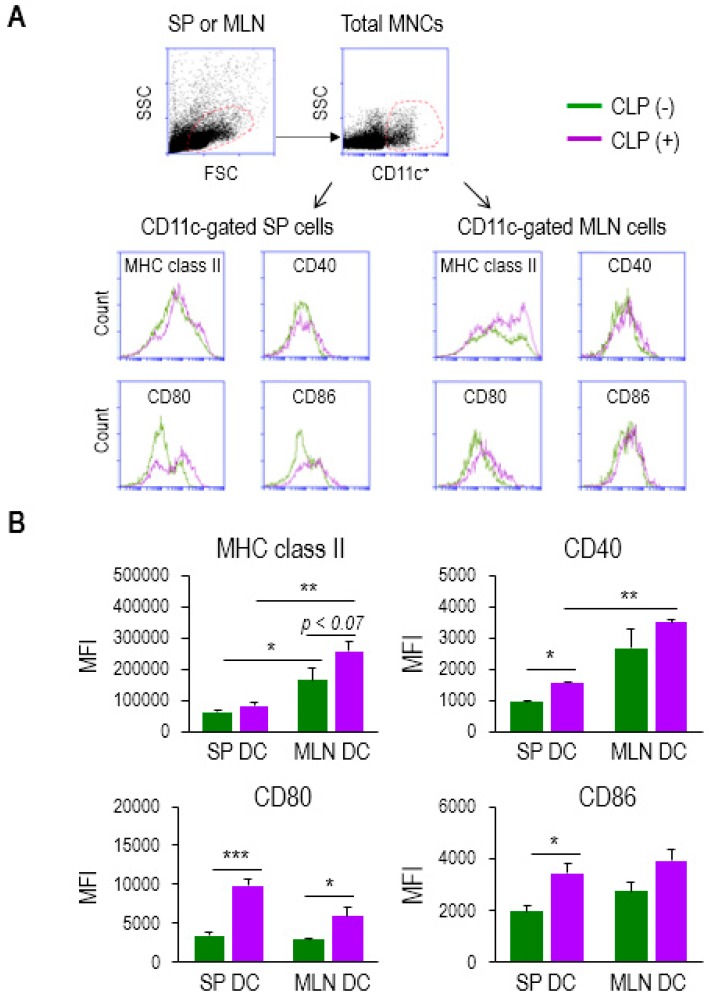
MLN DCs in sepsis elevated the surface expression of activation markers. (**A**) Mononuclear cells (MNCs) were isolated from single-cell suspensions of SP and MLNs. CD11c^+^ population was gated among total MNCs stained to determine the expression levels of APC markers including MHC class II, CD40, CD80, and CD86. Flow cytometry histograms show sepsis-induced change in their expression. (**B**) The bar graphs represent the mean ± SEM for MFI obtained from 4 to 5 mice per group, in which green and purple bars indicate without (−) and with (+) CLP, respectively. MFI, mean fluorescence intensity. Data are representative of at least three independent experiments. * 0.01 < *p* < 0.05, ** 0.01 < *p* < 0.001, *** *p* < 0.001.

**Figure 4 biomedicines-07-00052-f004:**
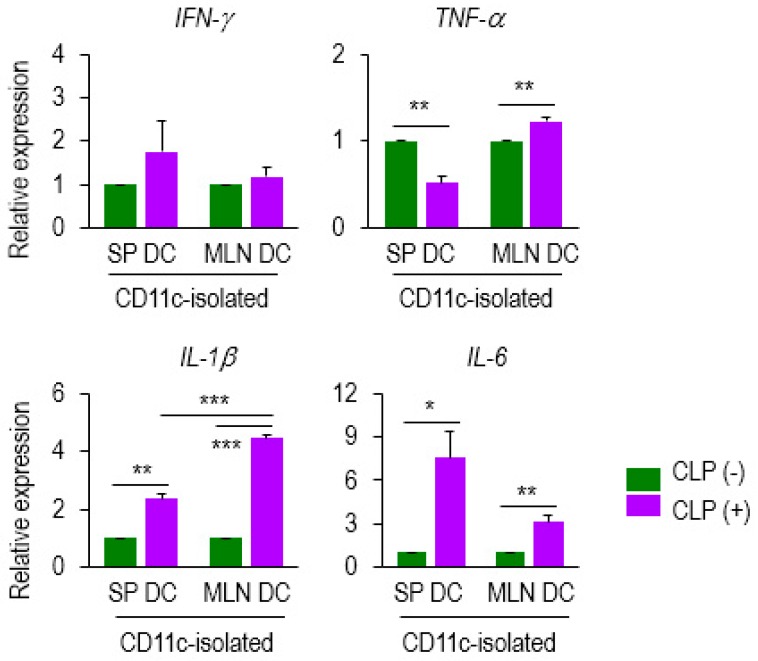
DC expression of several mediators was altered at the mRNA level during sepsis. Expression of several genes (*IFN-γ*, *TNF-α, IL-1β*, and *IL-6*) was examined on CD11c^+^ DCs isolated from SP DCs and MLN DCs of either CLP (−) or (+) mice. Data from qPCR analysis were normalized to those controls that represented samples from CLP (−) mice after being normalized to a reference β-actin gene (2^−ddCT^). Bar graphs represent the mean ± SEM obtained from 4 to 5 mice per group. Data are representative of at least three separate experiments. * 0.01 < *p* < 0.05, ** 0.01 < *p* < 0.001, *** *p* < 0.001.

**Figure 5 biomedicines-07-00052-f005:**
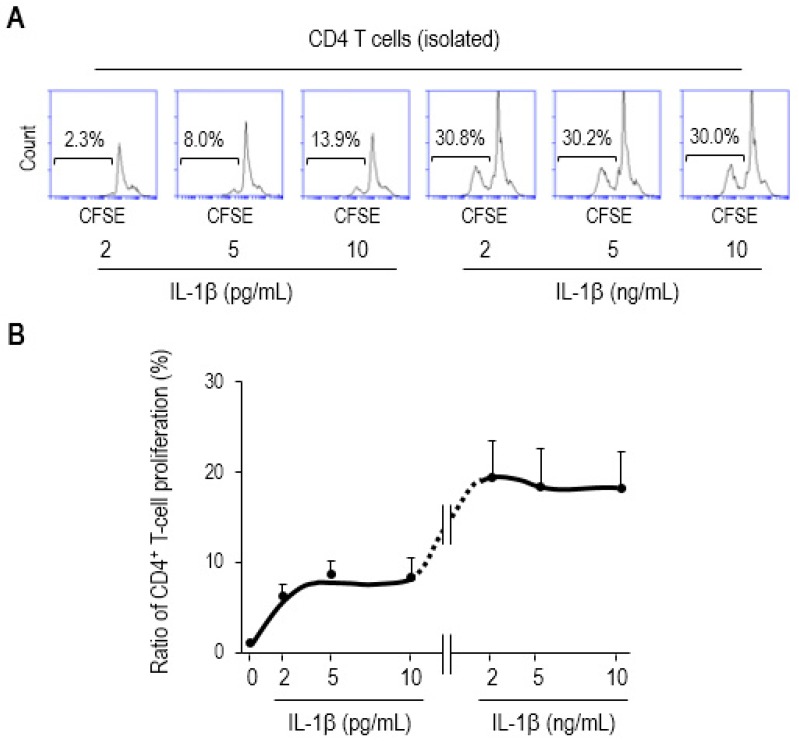
IL-1β augments CD4 T-cell proliferation. The CD4 T cells were isolated from single-cell suspensions of SP of Balb/c mice, fluorescently labeled with CFSE, treated with recombinant mouse IL-1β (rmIL-1β) at the indicated concentrations, and further incubated for 3 days. (**A**) Flow-cytometry histograms show representative results. The proliferation ratios were determined via measuring diluted fluorescent intensity of a histogram in flow cytometry in which the numbers inside squares represent the percentages of bracketed regions. (**B**) These results were shown as a line graph connected by the points that indicate different concentration of rmIL-1β used in the assay. The concentrations of exogenously treated IL-1β used in this study were similar to those in previous proof-of-the-principle studies involving ED_50_ (2–10 pg/mL) [45] (as also described in the datasheet provided by the manufacturer, R&D Systems). The 1000-times higher concentration (2–10 ng/mL) was included in this assay as well. Data are representative of at least two separate experiments.

**Figure 6 biomedicines-07-00052-f006:**
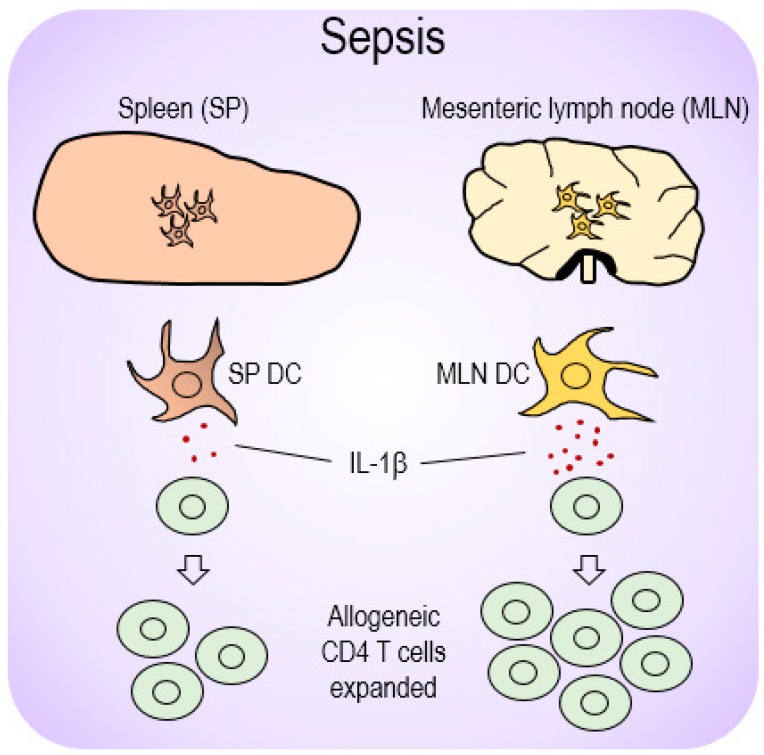
A proposed model for differential DC-mediated immune responses depending on location during sepsis. Sepsis initiated upregulation of the pro-inflammatory cytokine, IL-1β, from those DCs present in mucosal lymphoid organs (MLN DCs) and subsequently elevated allogeneic CD4 T cells during MLR. However, SP DCs that produced relatively low levels of IL-1β restricted CD4 T-cell proliferation, and thereby presumably caused an immunosuppressive effect during sepsis.

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
