# Peer review of "Differential Roles of Dendritic Cells in Expanding CD4 T Cells in Sepsis"

_biomedicines, 2019, doi:10.3390/biomedicines7030052_

Reviewer 1 Report

In this manuscript, Darkwah et al., have analyzed the differential roles of dendritic cells in expanding CD4+ T cells in sepsis. Authors found that mucosal DCs but not splenic DCs, from septic mice play a differential role in the activation or proliferation of CD4 T cells. This could be due to release of IL-1beta from mucosal DCs. Furthermore, they found that the activation markers are increased in MLN DCs in CLP mice.

This manuscript provides some interesting observations but needs further investigations. The experiments need to be shown in a more convincing way with appropriate controls. The bar graphs alone are not convincing enough to justify their conclusion. Further questions need to be addressed.

 Major Concerns

1.     In Figure 1, show flow cytometry plots based on which graphs were made. Also analyze inguinal lymph node as control to check increased DCs are specific to MLNs. 

2.    Previous studies have suggested that a profound loss in the number of CD11c+ DCs was observed in spleen after sepsis at the time ranging from 12 h to 3 days. Whereas, in the current study authors did not find any differences in SP CD11c+ DCs. It is important to mentionthe exact day of harvest. This will allow people in the field to know more about the moderate sepsis (mentioned in the current study) and lethal sepsis. If there is discrepancy between different time points it is worth mentioning that data and provide a comparison. 

3.     Figure 2 and 5, show CFSE staining plots and calculate MFI in addition to ratio of CD4+ T cells proliferation.

4.     It is difficult to understand the rationale to check for PD-L1 or PD-L2 expression on CD11c+ DCs. Checkpoint inhibitors are well studied in the case of tumor studies but not for sepsis. Better to provide a better rationale or remove as it is not adding any value to the manuscript.

5.     IL-1b has been shown previously to induce T cell proliferation. In the context of current manuscript, it would be interesting to show that this augmentation is specific to IL-1b but not IFNg or IL-6 or TNF.

Minor comments

1.     Provide a valid reason for excluding payer’s patches from further analysis. This should be discussed.

2.     It looks like Ifngexpression is increased in SP DCs in CLP+ mice, but it is not significant. Be careful and critical in performing statistical analysis. It is always advisable to mention exact sample size and the software used to perform the test (Prism or MS Excel, etc) with any corrections applied. 

Author Response

In this manuscript, Darkwah et al., have analyzed the differential roles of dendritic cells in expanding CD4+ T cells in sepsis. Authors found that mucosal DCs but not splenic DCs, from septic mice play a differential role in the activation or proliferation of CD4 T cells. This could be due to release of IL-1beta from mucosal DCs. Furthermore, they found that the activation markers are increased in MLN DCs in CLP mice.

This manuscript provides some interesting observations but needs further investigations. The experiments need to be shown in a more convincing way with appropriate controls. The bar graphs alone are not convincing enough to justify their conclusion. Further questions need to be addressed.

We appreciate Reviewer #1's very critical comments. We have addressed the comments of Reviewer #1 in a point-by-point manner as outlined below.

Major Concerns

 1.     In Figure 1, show flow cytometry plots based on which graphs were made. Also analyze inguinal lymph node as control to check increased DCs are specific to MLNs. 

We are grateful for these important suggestions. In Supplemental Figure 1 newly added in the revised manuscript, we have shown the represent flow-cytometry plots on which Figure 1 graphs were made. We have explained it in the text (lines 85-87). In addition, the PLN analyzed in this study was inguinal lymph nodes that the Reviewer suggested us to analyze. We have described it in the text as well as Figure 1 legend in the revised manuscript (lines 80, 82, and 94-95).

2.    Previous studies have suggested that a profound loss in the number of CD11c+ DCs was observed in spleen after sepsis at the time ranging from 12 h to 3 days. Whereas, in the current study authors did not find any differences in SP CD11c+ DCs. It is important to mention the exact day of harvest. This will allow people in the field to know more about the moderate sepsis (mentioned in the current study) and lethal sepsis. If there is discrepancy between different time points it is worth mentioning that data and provide a comparison.

We appreciate these very critical comments and suggestions made by Reviewer #1. We have indicated the time (“24 hour”) at which organ harvest is done after CLP surgery in the Methods section (4.2. Polymicrobial sepsis, lines 357-358). We have also explained levels of septic severity different depending on the ligation length used in CLP model in the same section (lines 354-356) of the revised manuscript. We also totally agree with and appreciate these important points indicated by Reviewer #1 that any discrepant outcomes depending on different time points need to be mentioned. It would be worthwhile to compare different conditions including varying levels of severity or duration to gain a more comprehensive overview in the context of a sepsis model. We have discussed these important points as a separate paragraph in Discussions section (lines 329-335) of the original as well as revised manuscript.

3.     Figure 2 and 5, show CFSE staining plots and calculate MFI in addition to ratio of CD4+ T cells proliferation.

In accordance with Reviewer #1's comments, we have revised Figure 2 and Figure 5 by newly adding Figure 2A and Figure 5A showing their representative CFSE staining plots. According to these newly added results, we have revised the text (lines 106-107 and 193-195) and legends (lines 119-121 for Figure 2; lines 198-204 for Figure 5) in the revised manuscript. Because the proliferation of cells is generally quantified by dilution of CFSE dye, we have shown the CD4+ T-cell proliferation as the diluted ratio, instead of MFI, in the histograms, which is displayed by the bar graphs in Figures 2 and 5.

4.     It is difficult to understand the rationale to check for PD-L1 or PD-L2 expression on CD11c+ DCs. Checkpoint inhibitors are well studied in the case of tumor studies but not for sepsis. Better to provide a better rationale or remove as it is not adding any value to the manuscript.

We completely agree with the suggestions of Reviewer #1 to remove the results of PD-L1 and PD-L2 expressions (Figure 4A). We have accordingly revised Figure 4 and some descriptions (2.3. Level of IL-1b mRNA is markedly increased in septic MLN DCs) in the text as well as the legend (lines 180-182).

5.     IL-1b has been shown previously to induce T cell proliferation. In the context of current manuscript, it would be interesting to show that this augmentation is specific to IL-1b but not IFNg or IL-6 or TNF.

We are thankful for these critical suggestions made by Reviewer #1. We have included the results obtained from analyzing proliferation of CD4 T cells treated with TNF-a in newly added Supplemental Figure 4. We were thus able to confirm that the augmentation of CD4 T cells is possibly specific to IL-1b but not to TNF-a. We have thus revised the text (lines 209-211) in the revised manuscript.

Minor comments

Provide a valid reason for excluding payer’s patches from further analysis. This should be discussed.

We are grateful for Reviewer #1’s comments. PP is a mucosal lymphoid tissue but known to be prone to be sepsis-induced apoptotic cellular loss (Ayala et al. Blood 1998, 91:1362-72; Hiramatsu et al. Shock 1997, 7:247-53). In accordance with these findings, the CLP mice exhibited a marked reduction in PP size (data not shown), which made it difficult to separate the DCs enough for performing analysis. Thus, MLN was used for providing mucosal DCs in current analyses. We have discussed these important points in the text (lines 99-102) and added those references mentioned above (refs. 32 and 33).

2.     It looks like Ifng expression is increased in SP DCs in CLP+ mice, but it is not significant. Be careful and critical in performing statistical analysis. It is always advisable to mention exact sample size and the software used to perform the test (Prism or MS Excel, etc) with any corrections applied. 

We appreciate these important points and suggestions. We have thus performed additional Q-PCR analysis using more mice to revise the results of ifn-g expression. In addition, we have also analyzed il-6 expression in those mice to validate it as well. We have revised Figure 4 and text (lines 170-173) in the revised manuscript.

Reviewer 2 Report

Darkwah et al, demonstrate the role of MLN –CD11c+ cells and IL-1β in CD4 T cells amplification and their role on sepsis. In general the experiments are performed well with suitable controls.  Authors use the term DCs to represent all CD11c+ cells that can even include some macrophages and as authors described in discussion they are heterogeneous population. One of the draw back of the study is the use of this heterogeneous population to reach conclusion that DC location is more important than the contribution of DC subsets.  When the authors are comparing sepsis model, ie CLP+ and CLP- conditions, as well as the tissue locations the following things should be considered.

·         Authors are overlooking the contribution of inflammatory dendritic cells, generally derived from monocytes. Studies very well demonstrated the differention and amplification of monocyte compartment during sepsis ( https://doi.org/10.3389/fimmu.2018.00823).

·         The author may be comparing two conditions were one have more of steady state DC and another one have a major contribution of inflammatory DCs.

·         Under inflammatory condition inf-DCs migrate to lymph node and not to spleen; so you may not observe any significant change in CD11c population in spleen but in lymph node.

·         In general a steady state DC and inf-DCs have a slightly different level of MHC II expression and we can clearly see the multiple peaks of MHC II in fig 3a.

·         Thus we can definitely assume the cell subsets in both the conditions are entirely different and their role on specific outcome is not well studied.

So I recommend the authors should perform a phenotype profiling of the CD11c+ cell in CLP+/- conditions.

The authors should perform a FACS staining on the CD11c cells to identify the cell subset and the increased frequency of inflammatory DCs in CLP+ conditions.  It will be great to do a FACs staining with, Lin neg, MHC II, CD11c, CD11b, CD24, CD26, XCR1, CD103/CD8, SIRPa, Siglec H, CD206, and other macrophage markers. There are publications described with minimum markers forthe detailed characterization of the subsets (PMID: 27637149).

The experiment will identify the role of specific subsets for the observed outcome and understand the biology.

Please analyse the CCR2 and CCR7 levels in these cells by FACS

Authors describe that prolonged immunocompromised condition in sepsis patients; does the IL-1 β induced CD4 T cell proliferation specifically skew these T cells to T reg?  It will be nice to see the FoxP3 expression on these T cells after expansion.

Minor comments

·         Figure 5: It will be great to represent the mean ±SD from at least three independent experiments

·         I hope the authors are using the term systemic DC for Steady state DC or lymphoid tissue resident DCs. The term steady state DCs may be more convincing than systemic DCs.

Author Response

Darkwah et al, demonstrate the role of MLN –CD11c+ cells and IL-1β in CD4 T cells amplification and their role on sepsis. In general the experiments are performed well with suitable controls.  Authors use the term DCs to represent all CD11c+ cells that can even include some macrophages and as authors described in discussion they are heterogeneous population. One of the draw back of the study is the use of this heterogeneous population to reach conclusion that DC location is more important than the contribution of DC subsets.  When the authors are comparing sepsis model, ie CLP+ and CLP- conditions, as well as the tissue locations the following things should be considered.

·         Authors are overlooking the contribution of inflammatory dendritic cells, generally derived from monocytes. Studies very well demonstrated the differention and amplification of monocyte compartment during sepsis ( https://doi.org/10.3389/fimmu.2018.00823).

·         The author may be comparing two conditions were one have more of steady state DC and another one have a major contribution of inflammatory DCs.

·         Under inflammatory condition inf-DCs migrate to lymph node and not to spleen; so you may not observe any significant change in CD11c population in spleen but in lymph node.

·         In general a steady state DC and inf-DCs have a slightly different level of MHC II expression and we can clearly see the multiple peaks of MHC II in fig 3a.

·         Thus we can definitely assume the cell subsets in both the conditions are entirely different and their role on specific outcome is not well studied.

So I recommend the authors should perform a phenotype profiling of the CD11c+ cell in CLP+/- conditions.

The authors should perform a FACS staining on the CD11c cells to identify the cell subset and the increased frequency of inflammatory DCs in CLP+ conditions.  It will be great to do a FACs staining with, Lin neg, MHC II, CD11c, CD11b, CD24, CD26, XCR1, CD103/CD8, SIRPa, Siglec H, CD206, and other macrophage markers. There are publications described with minimum markers for the detailed characterization of the subsets (PMID: 27637149).

The experiment will identify the role of specific subsets for the observed outcome and understand the biology.

We appreciate for these critical comments and suggestions made by Reviewer #2. We performed flow cytometry analysis to determine the sepsis-induced changes in expression levels of inflammatory DC markers. We first examined change in MHC class II expression on lineage marker-CD11c+ cells of SP and MLN tissues and found sepsis-induced DC augmentation of MHC class II expression both SP and MLN. We next investigated expression of inflammatory DC markers on lineage marker-CD11c+MHC class II+ cells of SP and MLN tissues of CLP (+) or CLP (-) mice. We have added the results in Supplemental Figures 2 and 3 and revised the text (lines 150-163). We have also added relevant references including those which the Reviewer kindly suggested (refs. 36-40).

Please analyse the CCR2 and CCR7 levels in these cells by FACS.

We have analyzed sepsis-induced change in CCR2 and CCR7 expressions of the lineage marker-CD11c+MHC-II+ cells from SP or MLN of CLP (-) or (+) mice. We have added the data in the Supplemental Figure 3B. We have placed the relevant descriptions in the revised manuscript (lines 163-166) and added two relevant references (refs. 41-43).

Authors describe that prolonged immunocompromised condition in sepsis patients; does the IL-1 β induced CD4 T cell proliferation specifically skew these T cells to T reg?  It will be nice to see the FoxP3 expression on these T cells after expansion.

We are grateful for the Reviewer #2’s very important comments. As suggested by the Reviewer, we have examined the effect of IL-1b on differentiation to Treg cells by measuring the FoxP3 expression and placed the results in the Supplemental Figure 5. We have added the relevant descriptions and a reference (ref. 45) in the revised manuscript (lines 212-218).

Minor comments

·         Figure 5: It will be great to represent the mean ±SD from at least three independent experiments

We appreciate for the Review’s important suggestion. We have revised the Figure 5 by replacing it the data obtained from combination of three to four independent experiments that indicate Figure 5B containing the mean values (dots) and ± SEM. We have revised the text (lines 193-195) and the Figure legends (lines 198-204) in the revised manuscript.       

I hope the authors are using the term systemic DC for Steady state DC or lymphoid tissue resident DCs. The term steady state DCs may be more convincing than systemic DCs.

We appreciate the Reviewer #2’s important comments. In current manuscript, we used the term “systemic DCs” in order to compare MLN (mucosal) DCs and SP (systemic) DCs under steady state (CLP -) and inflamed (CLP +) conditions. The “systemic DCs” meant the DCs isolated from SP which is a “systemic” tissue, whose nomenclature might have caused any confusion. The word “systemic DC” was used in this manuscript, to address the point of “systemic DCs compared to mucosal DCs”, instead of “systemic DCs compared to inflamed DCs”.

Round  2

Reviewer 1 Report

The authors have addressed all the concerns and suggestions. I have a few suggestions with data presentation and formatting:

1.    For mice, gene symbols are italicized, with only the first letter in upper-case (for ex., Il6TnfIl1b). Change it throughout the manuscript.

2.    Lineage marker terminology is misleading in this manuscript. (For ex., only 2 lymphoid and 1 myeloid lineage marker have been used). Instead of using the term lineage marker, use just the marker names. Make it consistent throughout the manuscript.

3.    It is always advisable to mention the program/software used to perform statistical analysis (for ex., MS Excel, GraphPad Prism, etc).

4.    Mention the clones for all the flow antibodies used in addition to the makers.

5.    Supplemental figure 1 shows 2 graphs each for CLP (-) and CLP (+). Are these from two different mice? Specify in the legend. It is mentioned in the Figure legend that, “Data are representative of at least three independent experiments that show similar results”. Either show all 3 or just show 1 and modify the Figure and legend accordingly.

 Author Response

 Reviewer #1

The authors have addressed all the concerns and suggestions. I have a few suggestions with data presentation and formatting:

We are thankful for the positive comments of Reviewer #1' on our revised manuscript. We have revised our manuscript based on the additional suggestions of Reviewer #1 in a point-by-point manner as outlined below.

For mice, gene symbols are italicized, with only the first letter in upper-case (for ex., Il6TnfIl1b). Change it throughout the manuscript.

We are thankful for Reviewer #1’s suggestions. We have corrected the gene symbols by changing their first letters to uppercase throughout the manuscript including the text (lines 194-196), Figure 4 and Figure 4 legend (line 183) in the revised manuscript.

Lineage marker terminology is misleading in this manuscript. (For ex., only 2 lymphoid and 1 myeloid lineage marker have been used). Instead of using the term lineage marker, use just the marker names. Make it consistent throughout the manuscript.

We appreciate for Reviewer #1’s comments. We completely agree with the suggestion of Reviewer #1 to use marker names instead of the term lineage marker. We have revised to make it all consistent throughout the manuscript including the text (lines 151-152, 158, 160, 163, 170, 172, etc.), Methods (4.5. Flow cytometry analysis section), Supplemental Figure 2, Supplemental Figure 3, Supplemental Figure 2 legend, and Supplemental Figure 3 legend in the revised manuscript.

 It is always advisable to mention the program/software used to perform statistical analysis (for ex., MS Excel, GraphPad Prism, etc).

We have described that statistical analysis was done using Microsoft Excel in the Methods in the revised manuscript (lines 433-434).

Mention the clones for all the flow antibodies used in addition to the makers.

We have described the clone names for all the antibodies used in the flow cytometry analysis in the Methods in the revised manuscript (lines 402-413).

Supplemental figure 1 shows 2 graphs each for CLP (-) and CLP (+). Are these from two different mice? Specify in the legend. It is mentioned in the Figure legend that, “Data are representative of at least three independent experiments that show similar results”. Either show all 3 or just show 1 and modify the Figure and legend accordingly.

We are very grateful for the indication made by the Reviewer #1. We have revised the Supplemental Figure 1 by correcting the indications of CLP (-) and CLP (+). As also indicated in the Figure legend, Supplemental Figure 1 shows the dot-plots obtained from each representative animal out of three CLP (-) or CLP (+) mice.

Reviewer 2 Report

Darkwah et al., demonstrate the differential roles of dendritic cells in expanding CD4 T cells in sepsis and DCs modulate the expansion through IL1β. The authors included additional information based on the reviewers comment. The new additions definitely improved the manuscript and there are few additional points require some clarifications. The main concerns are on the FACS staining depicted in supplementary figure 1, 2 and 3.  It is not very clear on the gating strategy why the authors consider these cells are Clas II+ CD11c+. If you look at the other published literature (For eg : - PMID: 26938654 or PMID: 27637149 )  we can very clearly see the Class II positive cells or CD11c positive cells. Even if you look at the Figure 3A, the CD11c+ cells are distinctive clusters where as in supplementary figure 1, the population is very vague and even the representative figure in supplementary have low frequency of CD11c+ cells in MLN in CLP+ condition. One of the major idea for staining with multiple inflammatory markers are to distinguish the mononuclear phagocytes to subsets and understand the specific contribution of the subsets. The contribution of the specific subsets to induce the proposed outcome is very important information for the field.  The experimental efforts of the authors are not really reflecting in the FACS analysis part and it will be better have reanalysis of the FACS data to get better figures for a rigorous reporting. It will be great to also provide the % positive cells of each subsets or markers including CD11b+, CD11c+, F4/80 or CD206+ etc to estimate the frequency of different cell subset in each condition. Please use a gating strategy described in any of the above papers to identify the subsets. Does the FACS staining shows any specific CD206+ CD11c+ subsets in CLP+ condition or any other subsets in your experimental conditions? Does it related to the specific cytometer used for the experiments? Please provide a flow plots with gate from the software than manually applying gates in the cropped dot plots. If you are considering Lineage negative CD11c+Class II + cells are DC, please use the total cells for calculating the MFI than a portion of the cells in the gate. Providing the antibody clones and conjugates used for the experiments may be useful for future references and reproduce the results by other researchers.

Author Response

Reviewer #2

Darkwah et al., demonstrate the differential roles of dendritic cells in expanding CD4 T cells in sepsis and DCs modulate the expansion through IL1β. The authors included additional information based on the reviewers comment. The new additions definitely improved the manuscript and there are few additional points require some clarifications. The main concerns are on the FACS staining depicted in supplementary figure 1, 2 and 3.  It is not very clear on the gating strategy why the authors consider these cells are Clas II+ CD11c+. If you look at the other published literature (For eg : - PMID: 26938654 or PMID: 27637149 )  we can very clearly see the Class II positive cells or CD11c positive cells. Even if you look at the Figure 3A, the CD11c+ cells are distinctive clusters where as in supplementary figure 1, the population is very vague and even the representative figure in supplementary have low frequency of CD11c+ cells in MLN in CLP+ condition. One of the major idea for staining with multiple inflammatory markers are to distinguish the mononuclear phagocytes to subsets and understand the specific contribution of the subsets. The contribution of the specific subsets to induce the proposed outcome is very important information for the field.  The experimental efforts of the authors are not really reflecting in the FACS analysis part and it will be better have reanalysis of the FACS data to get better figures for a rigorous reporting. It will be great to also provide the % positive cells of each subsets or markers including CD11b+, CD11c+, F4/80 or CD206+ etc to estimate the frequency of different cell subset in each condition. Please use a gating strategy described in any of the above papers to identify the subsets. Does the FACS staining shows any specific CD206+ CD11c+ subsets in CLP+ condition or any other subsets in your experimental conditions? Does it related to the specific cytometer used for the experiments? Please provide a flow plots with gate from the software than manually applying gates in the cropped dot plots. If you are considering Lineage negative CD11c+Class II + cells are DC, please use the total cells for calculating the MFI than a portion of the cells in the gate. Providing the antibody clones and conjugates used for the experiments may be useful for future references and reproduce the results by other researchers.

We appreciate for the valuable comments made by Reviewer #2. We have revised our manuscript based on those critical suggestions of Reviewer #2 as outlined below.

We have revised the Figures (including Figure 3A, Supplemental Figure 2A, and Supplemental Figure 3A) by providing cropped dot plots for gating in the flow cytometry software. Moreover, we have revised Supplemental Figure 3 by analyzing expressions of inflammatory DC markers on total CD3e-B220-Gr-1-TER119- (lineage-marker negative) cells and adding those results to Supplemental Figure 3B. We have fixed properly the original Supplemental Figure 3B and renamed it to Supplemental Figure 3C. We have revised the text (lines 160-167) and Supplemental Figure 3 legend in the revised manuscript. We have added relevant descriptions for the results of CCR2 and CCR7 expressions in the text (lines 174-179) of the revised manuscript.

Regarding the critical indication that the CD11c+MHC class II+ cells shown in flow cytometry is vague and somewhat less distinctive in the Supplemental Figure 3A, we agree with these valuable points. The CD11c+ cells gated in dot plot of Supplemental Figure 3A are different, in quantity and quality, from those shown in Figure 3A. It might be, at least partly, because the dot plot of Supplemental Figure 3A represents CD11c-expressing cells gated out of CD3e-B220-Gr-1-TER119- (lineage-marker negative) cells, while the dot plot in Figure 3A shows CD11c-expressing cells gated out of total cells, although it still needs more valid explanations for those discrepant patterns. We have added a brief explanation for this note (lines 161-162) along with two relevant references (refs. 38 and 41) suggested by Reviewer #2. We have also revised the dot plots in Supplemental Figure 3A.

We are thankful for the critical comments on importance of elucidating correlation between distinct DC subsets and their functional outcome in inflammation. We agree with these informative suggestions and have addressed a description in the text (lines 340-341) of the revised manuscript.

We have analyzed the percentage of inflammatory DC-marker expressing cells in total CD3e-B220-Gr-1-TER119-CD11c+MHC class II+ cells and newly added those data to Supplemental Figure 4. We have revised the text (lines 168-173) by adding relevant descriptions in the revised manuscript.

Regarding the FACS analysis to show whether any inflammatory marker-expressing distinct CD11c+ subsets are induced by sepsis, a remarkable increase in MFI and positive-cell ratio of F4/80 was shown in SP DCs in sepsis (Supplemental Figure 3C and Supplemental Figure 4). Thus, we have revised the text (lines 342-350) by explaining the results and speculation as a paragraph in Discussions about the possible induction of monocyte-derived F4/80+/CD3e-B220-Gr-1-TER119-CD11c+MHC class II+ cells. We have added a relevant reference (ref. 79) in the revised manuscript (line 346).

We have revised the Methods section (4.5. Flow cytometry analysis) by providing clones and conjugates of all the antibodies used in the experiments (lines 402-413) in the revised manuscript.